# Design, Synthesis and Investigation of the Potential Anti-Inflammatory Activity of 7-*O*-Amide Hesperetin Derivatives

**DOI:** 10.3390/molecules24203663

**Published:** 2019-10-11

**Authors:** Yilong Zhang, Yan Zheng, Wen Shi, Yahui Guo, Tao Xu, Zeng Li, Cheng Huang, Jun Li

**Affiliations:** 1The Key Laboratory of Major Autoimmune Diseases, Anhui Province, Anhui Institute of Innovative Drugs, School of Pharmacy, Anhui Medical University, Hefei 230032, China; zhangyl_ahmu@126.com (Y.Z.); 13170023235@163.com (Y.Z.); shiwenahaq@163.com (W.S.); yahuiguo_1@163.com (Y.G.); xutao@ahmu.edu.cn (T.X.); lizeng@ahmu.edu.cn (Z.L.); 2The Key laboratory of Anti-Inflammatory Immune Medicines, Ministry of Education, Hefei 230032, China; 3Institute for Liver Diseases of Anhui Medical University, Anhui Medical University, Hefei 230032, China

**Keywords:** hesperetin derivatives, synthesis, inflammatory, NF-κB, structure-activity relationships.

## Abstract

To develop new anti-inflammatory agents, a series of 7-*O*-amide hesperetin derivatives was designed, synthesized and evaluated for anti-inflammatory activity using RAW264.7 cells. All compounds showed inhibitory effect on LPS-induced NO production. Among them, 7-*O*-(2-(Propylamino)-2-oxoethyl)hesperetin (**4d**) and 7-*O*-(2-(Cyclopentylamino)-2-oxoethyl)hesperetin (**4k**) with hydrophobic side chains exhibited the most potent NO inhibitory activity (IC_50_ = 19.32 and 16.63 μM, respectively), showing stronger inhibitory effect on the production of pro- inflammatory cytokines tumor necrosis factor (TNF-α), interleukin-6 (IL-6) and interleukin-1β (IL-1β) than indomethacin and celecoxib at 10 μM. The structure-activity relationships (SARs) suggested that the 7-*O*-amide unit was buried in a medium-sized hydrophobic cavity of the bound receptor. Furthermore, compound **4d** could also significantly suppress the expression of inducible nitric oxide synthase enzymes (iNOS) and cyclooxygenase-2 (COX-2), through the nuclear factor-kappa B (NF-κB) signaling pathway.

## 1. Introduction

Inflammation which is a biological response to harmful stimuli such as pathogens that cause tissue and cell damage is a central feature of many pathological conditions [1,2]. Persistent inflammation can cause severe damage to organisms, which is detrimental to health [3]. Citrus is the fruit of Rutaceae plants that produce oranges (*Citrus reticulata* Blanco) and its cultivars. Hesperetin and hesperidin are the primary active contents of citrus fruits. Hesperidin is a flavanone glycoside, which is abundantly present in citrus fruits and often consumed in the form of orange juice. Hesperetin, derived from the hydrolysis of hesperidin, is the major flavonoid occurring in sweet oranges and orange juice [4,5]. Previous studies have shown that hesperetin has a variety of pharmacological effects, including anti-inflammatory [6,7], anti-tumor [8], anti-oxidant [9] and neuroprotective properties [10,11,12,13]. Although hesperetin has a wide range of pharmacological effects, its poor aqueous solubility, low bioavailability, rapid clearance in vivo, and short plasma circulation time limit its wide application [14,15]. Structural modification of bioactive natural products according to molecular characteristics was an essential approach in the search for new lead compounds. 

In our previous studies, we synthesized a series of hesperidin derivatives and screened their anti-inflammatory activities in vitro as well as in vivo [16,17,18,19]. An earlier study reported that the elimination half-life (t_1/2_) of hesperetin was 1.78 h in rats [15] and 3.12 h in healthy human subjects [14]. Pharmacokinetic studies of Mannich base hesperetin derivatives have shown that were rapidly metabolized, primarily to different combinations of 7-*O*-glucuronides and 7-*O*-sulfates [19,20]. We have designed and synthesized 7-*O*-substituent hesperetin derivatives for improving the activity and bioavailability [16,21,22]. It was found that the introduction of hydrophobic alkyl and benzyl groups at 7-OH improved their anti-inflammatory activities [22]. However, the introduction of hydrophobic α,β-unsaturated amides resulted in weak anti-inflammatory activity [16]. We considered that the size and shape of the cavity of bound receptor, which in the 7-OH direction, affected the anti-inflammatory activity of 7-*O*-substituent hesperetin derivatives. Based on the above observations, we proceeded with modification of 7-*O*-substituent hesperetin derivatives using following strategies: (a) shortening the length of α,β-unsaturated acid moiety at 7-OH, namely replacing it with acetic acid (**3**); (b) converting the carboxyl group to an amide functionality with linear alkyl, branched alkyl, cycloalkyl, heterocycle or hydrophilic groups (Scheme 1). The inhibitory activity on NO, tumor necrosis factor (TNF-α), interleukin-6 (IL-6) and interleukin-1β (IL-1β) production in lipopolysaccharide (LPS)-induced mouse macrophage, abelson murine leukemia virus transformed (RAW264.7 cells) and anti-inflammatory mechanism of these derivatives investigated as well. 

## 2. Results and Discussion

### 2.1. Chemistry

Since hesperidin was extremely difficult to dissolve in common solvents and was relatively easily isolated from plants’ extract, the target compounds **4a**–**4l**, **5a**–**5b**, **6a**–**6f,** and **7a**–**7c** were synthesized starting from commercially available hesperidin as illustrated in Scheme 1. At the very start, hesperetin (**1**) was obtained by sulfuric acid hydrolysis of hesperidin in ethanol [23] and then the synthesis of the target intermediates was conducted. Compound **2** was prepared by a substitution reaction treating hesperetin with ethyl bromoacetate and K_2_CO_3_ [16,24]. As shown in Table 1, ethyl bromoacetate selectively substituted at the 7-OH position. Since compounds **2** and **3** were easily oxidized under pH > 10, hydrolysis reaction was carried out at low temperature as soon as possible. In the condensation reactions of amino and carboxyl groups, 1-(3-dimethyl- aminopropyl)-3-ethylcarbodiimide hydrochloride (EDC·HCl) and 1-hydroxybenzotriazole (HOBT) are indispensable catalysts [24,25]. ^1^H-NMR spectra and chemical shifts of hesperidin are shown in Appendix A of the Supporting Information.

### 2.2. Evaluation of Biological Activities

#### 2.2.1. Cytotoxicity of Hesperetin Derivatives

Compounds **4a**–**4l**, **5a**–**5b**, **6a**–**6f**, **7a**–**7c** were assessed for toxicity against RAW264.7 cell lines using the 3-(4,5-dimethylthiazol-2-yl)-2,5-diphenyltetrazolium bromide (MTT) assay [26]. The cytotoxicity tests were performed at the concentration of 40 μM and the results are shown in Table 2. After 40 μM treatment, the cell viability of all of the hesperetin derivatives was more than 90%. It was thus shown that the anti-inflammatory activities of hesperetin derivatives were not due to their cytotoxicity. In subsequent anti-inflammation studies, the highest concentration of hesperetin derivatives was set below 40 µM. The viability assay of RAW264.7 cells treated with different concentrations of compound **4d** are shown in Figure 1. There was no significant cytotoxicity of compound **4d** at concentrations up to 100 μM.

#### 2.2.2. Inhibitory Activity on LPS-Induced NO Production

NO is an important inflammatory mediator and is overproduced and secreted out of mouse macrophages in response to bacterial lipopolysaccharide [27]. To evaluate the anti-inflammatory activity of hesperetin derivatives, the secretion level of NO was detected in the medium of RAW264.7 cells after LPS stimulated with Griess reagent. Before the cells were treated with LPS (1 μg/mL) for 24 h, pretreatment with the compounds for 1 hour. Later, the expression level of NO production was markedly reduced. The inhibitory activity of the compounds as summarized in Table 3. All hesperetin derivatives could decrease the release of NO induced by LPS. Among them, compounds **4d**, **4e**, **4g**, **4k**, and **6a** significantly alleviated the increase of LPS-induced NO release. As expected, compounds with hydrophobic side chains exhibited better NO inhibitory activity than with hydrophilic side chains, such as **4d** > **7a**; **4g** > **7b**; **6a** > **6b**,**6c**. While extended the length of the amide side chain and the inhibitory activities increased (**4d** > **4c** > **4a**; **4e** > **4b**). These showed the side chains of 7-*O*-amide buried in a hydrophobic cavity. On the other hand, the cavity was medium-sized, large enough to accommodate cyclopentyl and propyl groups, but not enough to accommodate cyclohexyl and butyl ones (**4a** < **4b** < **4d** > **4f**; **4k** > **4l**). Overall, 7-*O*-amide hesperetin derivatives displayed preferably anti-inflammatory activity, particularly, compounds **4d** and **4k** exhibited the best inhibitory effect. At the same time, compound **4d** inhibited the production of NO in 38.17% at the concentration of 10 μM and the IC_50_ value of compound **4d** on NO inhibition was 19.32 μM while indomethacin and celecoxib showed values of 35.30 μM and 26.35 μM. These results showed that compound **4d** exerted the suppression of NO activity in a dose-dependent manner.

#### 2.2.3. Inhibitory Activity on TNF-α, IL-6 and IL-1β Production in LPS-Induced RAW264.7 Cells

When inflammation occurs, LPS-activated macrophages produce a variety of inflammatory cytokines, including TNF-α, interleukins (ILs), and nitric oxide (NO) [28], which show a host shielding effect and also accelerate the occurrence of the inflammatory response during inflammatory situations. To evaluate the anti-inflammatory activity of hesperetin derivatives, the levels of TNF-α, IL-6 and IL-1β in the medium of LPS-induced macrophage cells were detected by enzyme linked immunosorbent assay (ELISA). As shown in Figure 2A–C, the expression levels of TNF-α, IL-6 and IL-1β were significantly increased after treatment with LPS, compared to the Control group, and were significantly down-regulated by most of the hesperetin derivatives at the concentration of 10 μM, which was consistent with our observations of its activity in NO inhibition. Compound **4d**, **4e,** and **4k** had greater inhibitory effects than indomethacin (Ind) and celecoxib (Cel). Compound **4d** inhibited the LPS-induced production of IL-6, IL-1β and TNF-α in a dose-dependent manner (Figure 2D–F). Later, we will select one of them as lead compound and modify the structure at the 3’-OH, 5-OH and 4-carbonyl positions. Lipinski’s drug-likeness Rule of five (Ro5) suggests molecular weight (MW) <500 [29]. Compounds 4d (MW = 401) and 4k (MW = 427) have similar structures and should have the same mechanism. Therefore, compound **4d** with lower molecular weight was selected to further explore the mechanism(s) of the anti-inflammatory effect.

#### 2.2.4. Inhibitory Activity on Expression Levels of COX-2 and iNOS in LPS-Induced RAW264.7 Cells by Compound **4d**

In the process of inflammation, inducible nitric oxide synthase enzymes (iNOS) are induced to overexpression in inflammatory cells when exposed to pro-inflammatory cytokines, which further leads to excessive production of NO. The inhibition of NO production by compound **4d** may be correlated with its suppressive effect on LPS-induced iNOS. Thus, the inhibitory effects of compound **4d** on LPS-mediated expressions of iNOS were analyzed by western blotting [30,31]. Nitric oxide synthase (iNOS) protein levels were analyzed after exposure to LPS for 24 h in the presence or absence of compound **4d** (5, 10, 20 μM). As expected, LPS stimulation could markedly increase iNOS protein expression, and compound **4d** significantly decreased the expression level of iNOS induced by LPS in a dose-dependent manner (Figure 3). Cyclooxygenase-2 (COX-2) is an important indicator of anti-inflammatory inhibitors, maintaining a low level in normal tissues. COX-2 is rapidly up-regulated when inflammation occurs, and the downstream inflammatory mediators such as prostaglandin secretion rise rapidly, further aggravating the development of inflammation. We also found that compound **4d** was capable of reducing the expression level of COX-2 in a concentration-dependent way (Figure 3). Bay 11-7082 (5 μM) used as a positive control.

#### 2.2.5. Negative Regulation of NF-κB Signal Pathway in LPS-Induced RAW264.7 Cells by Compound **4d**

Stimulation with LPS leads to the activation of nuclear factor-kappa B (NF-κB), which is a well-known transcription factor that positively regulates inflammation. When NF-κB activation is stimulated by LPS, inhibitor of NF-κB (IκB) protein phosphorylated and degraded frees NF-κB p65 subunit from sequestration, allowing it to translocate to the nucleus, bind to target promoters, and turn on the transcriptions of inflammation genes including TNF-α and IL-6. Therefore, Western blotting was used to examine the effects of compound **4d** on LPS induced transcriptional activity of NF-κB in RAW264.7 cells. 

As shown in Figure 4, compound **4d** markedly decreased the phosphorylation of IκBα. The protein expressions of NF-κB and IκBα were also down-regulated, compared with the LPS induced group. At the same time, LPS increased NF-κB p65 nuclear translocation, whereas pretreatment with compound **4d** (5, 10, 20 µM) concentration-dependently reversed LPS-induced NF-κB p65 nuclear translocation, suggesting that compound **4d** inhibited NF-κB activation. These results further confirmed the anti-inflammatory effect of **4d**, at least in part, through modulation of NF-κB activation.

## 3. Materials and Methods

### 3.1. Chemistry

#### 3.1.1. General Information

All reagents were purchased from commercial sources and were used without further purification. Melting points were determined on an SGW^®^ X-4A apparatus (Shanghai Instrument Physical Optics Instrument Co., Ltd., Shanghai, China). ^1^H-NMR and ^13^C-NMR spectra were recorded on an AV-400 MHz instrument (Bruker, Fallanden, Switzerland) in dimethyl sulfoxide (DMSO-*d*_6_). Chemical shifts are reported in parts per million (δ) downfield from the signal of tetramethylsilane (TMS) as internal standards. Coupling constants are reported in Hz. The multiplicity is defined by s (singlet), d (doublet), t (triplet), br (broad) or m (multiplet). High resolution mass spectra (HRMS) were obtained on a 1260–6221 TOF mass spectrometry system (Agilent, Santa Clara, CA, USA). Column and thin-layer chromatography (CC and TLC, resp.) were performed on silica gel (200–300 mesh) and silica gel GF254 (Qingdao Marine Chemical Factory, Qingdao, China), respectively.

#### 3.1.2. Synthetic Methods for All Compounds

##### Synthesis of Hesperetin (**1**)

To a solution of ethanol (640 mL) and 98% H_2_SO_4_ (80 mL) was added hesperidin (72 g, 0.12 mol), the suspension was heated at 80 °C for 8 h. The reaction mixture was allowed to cool to room temperature, then poured into ice water (2 L). The precipitate was filtered, washed with water and dried. The mixture was dissolved in ethanol and heated to reflux for 1 h, heating was stopped and activated charcoal was added to the solution for 15 min. The solution was filtered and washed with hot ethanol. The ethanol solution was concentrated under the reduced pressure, and the residue was recrystallized from ethanol and CH_2_Cl_2_ to obtain hesperetin (**1**) [23] as a white powder (31.5 g, yield 87%), m.p. 229.5–231.6 °C; ^1^H-NMR: δ 12.14 (s, 1H, 5-OH), 10.80 (s, 1H, 7-OH), 9.11 (s, 1H, 3’-OH), 6.95–6.92 (m, *J* = 5.1, 3.2 Hz, 2H, 2’-H, 5’-H), 6.87 (dd, *J* = 8.3, 2.0 Hz, 1H, 6’-H), 5.89 (d, *J* = 2.1 Hz, 1H, 8-H), 5.88 (d, *J* = 2.1 Hz, 1H, 6-H), 5.43 (dd, *J* = 12.4, 3.0 Hz, 1H, 2-H), 3.77 (s, 3H, OCH_3_), 3.20 (dd, *J* = 17.1, 12.4 Hz, 1H, 3-H), 2.70 (dd, *J* = 17.1, 3.1 Hz, 1H, 3-H). ^13^C-NMR: δ 196.61, 167.22, 163.92, 163.24, 148.34, 146.85, 131.54, 118.20, 114.45, 112.34, 102.21, 96.31, 95.49, 78.68, 56.07, 42.46. HRMS (ESI): Calcd. C_16_H_14_O_6_, [M + H]^+^
*m*/*z*: 303.0875, found: 303.0881.

##### Synthesis of 7-*O*-(2-ethoxy-2-oxoethyl)hesperetin (**2**)

To a solution of hesperetin **1** (30.2 g, 100 mmol) and KHCO_3_ (30 g, 300 mmol) in DMF (400 mL) was added K_2_CO_3_ (12.4 g, 150 mmol) stirring at room temperature (25 °C) for 30 min, then bromoacetate (41.8 g, 250 mmol) was added to the reaction solution with vigorous stirring for 1 h. The reaction was monitored by TLC. The reaction mixture was acidified with dilute HCl to pH 5–6 and extracted with EtOAc. The organic phase was washed with brine solution for five times, then dried over anhydrous Na_2_SO_4_, filtered, and concentrated under the reduced pressure. The residue was recrystallized from ethanol and CH_2_Cl_2_ to obtain a white crude product. The crude product was recrystallized from CH_2_Cl_2_ and ether to obtain 7-*O*-(2-ethoxy-2-oxoethyl)hesperetin (**2**) [24] (white crystalline powder, 19.8 g, yield 50% for this reaction step, Scheme 1), m.p. 127.8–130.2 °C; ^1^H-NMR: δ 12.08 (s, 1H, 5-OH), 9.13 (s, 1H, 3’-OH), 6.98–6.91 (m, 2H, 2’-H, 5’-H), 6.89 (dd, *J* = 8.4, 1.5 Hz, 1H, 6’-H), 6.11 (d, *J* = 2.1 Hz, 1H, 8-H), 6.09 (d, *J* = 2.1 Hz, 1H, 6-H), 5.49 (dd, *J* = 12.5, 2.7 Hz, 1H, 2-H), 4.87 (s, 2H, ArOCH_2_C=O), 4.17 (q, *J* = 7.1 Hz, 2H, CCH_2_OC=O), 3.78 (s, 3H, OCH_3_), 3.28 (dd, *J* = 17.1, 12.6 Hz, 1H, 3-H), 2.75 (dd, *J* = 17.1, 2.9 Hz, 1H, 3-H), 1.21 (t, *J* = 7.1 Hz, 3H, CH_3_COC=O). ^13^C-NMR: δ 197.43, 168.50, 166.08, 163.54, 163.18, 148.43, 146.94, 131.35, 118.24, 114.59, 112.37, 103.46, 95.65, 94.74, 78.98, 65.22, 61.32, 56.10, 42.60, 14.48. HRMS (ESI): Calcd. C_20_H_20_O_8_, [M + H]^+^
*m*/*z*: 389.1252, found: 389.1257.

##### Synthesis of 7-*O*-(carboxymethyl)hesperetin (**3**)

Substituted derivative **2** (19.4 g, 50 mmol) was mixed with 10% ethanol aqueous solution (100 mL) under ice-cooling for 30 min. Cooled 10% NaOH (100 mL) was dropped to the reaction solution over 30 min at 0 °C, and the mixture was stirred for 30 min at the same temperature. The reaction mixture was acidified with dilute HCl (ice bath) to pH 3–4. The precipitate was filtered and washed with acidified water, The solid dried in vacuum to obtain the intermediate **3** as a yellow solid (16.5 g, 92% Scheme 1), m.p. 198.3–202.4 °C; ^1^H-NMR: δ 12.07 (s, 1H, 5-OH), 9.21 (s, 1H, 3’-OH), 6.97–6.92 (m, 2H, 2’-H, 5’-H), 6.89 (dd, *J* = 8.4, 1.8 Hz, 1H, 6’-H), 6.07 (d, *J* = 2.2 Hz, 1H, 8-H), 6.06 (d, *J* = 2.2 Hz, 1H, 6-H), 5.48 (dd, *J* = 12.5, 2.9 Hz, 1H, 2-H), 4.76 (s, 2H, ArOCH_2_C=O), 3.77 (s, 3H, OCH_3_), 3.26 (dd, *J* = 17.1, 12.5 Hz, 1H, 3-H), 2.75 (dd, *J* = 17.1, 3.0 Hz, 1H, 3-H). ^13^C-NMR: δ 197.33, 169.97, 166.25, 163.49, 163.11, 148.41, 146.86, 131.30, 118.29, 114.49, 112.34, 103.34, 95.59, 94.69, 78.93, 65.11, 56.08, 42.53. HRMS (ESI): Calcd. C_18_H_16_O_8_, [M + H]^+^
*m*/*z*: 361.0919, found: 361.0933.

#### 3.1.3. General Procedure for the Synthesis of Compounds **4a**–**l**, **5a**–**b**, **6a**–**g**, **7a**–**c**

To a solution of 7-*O*-(carboxymethyl)hesperetin (**3**, 720 mg, 2 mmol) in anhydrous CHCl_3_ without alcohol (40 mL) was added HOBt (811 mg, 6 mmol), EDC·HCl (780 mg, 4 mmol) and NEt_3_ (4 mmol, 0.4 mL), stirred at room temperature for 1 h. Then the corresponding amine (6.0 mmol, 3 equiv.) was added to the reaction solution with stirring for 3 h. The reaction mixture was acidified with dilute HCl to pH 7–8 and extracted with EtOAc. The organic phase was washed with saturated sodium bicarbonate solution for twice, then dried over anhydrous Na_2_SO_4_, filtered, and concentrated under the reduced pressure. The residue was recrystallized from ethanol and CH_2_Cl_2_ to obtain the amides **4a**–**l**, **5a**–**b**, **6a**, **6f**. The residues of **6b**–**e**, **7a**–**c** were purified by flash column chromatography (CHCl_3_/EtOAc = 5/1, *v*/*v*) [25] (white crystals or powder, yields 36–80% for this reaction step, Scheme 1).

##### 7-*O*-(2-(Methylamino)-2-oxoethyl)hesperetin (**4a**)

White crystals, 72% yield, m.p. 179.8–182.7 °C; ^1^H-NMR: δ 12.08 (s, 1H, 5-OH), 9.13 (s, 1H, 3’-OH), 8.07 (d, *J* = 4.5 Hz, 1H, NH), 6.97–6.92 (m, 2H, 2’-H, 5’-H), 6.88 (dd, *J* = 8.4, 1.8 Hz, 1H, 6’-H), 6.11 (d, *J* = 2.2 Hz, 1H, 8-H), 6.10 (d, *J* = 2.2 Hz, 1H, 6-H), 5.49 (dd, *J* = 12.5, 2.9 Hz, 1H, 2-H), 4.54 (s, 2H, ArOCH_2_C=O), 3.78 (s, 3H, OCH_3_), 3.27 (dd, *J* = 17.1, 12.5 Hz, 1H, 3-H), 2.75 (dd, *J* = 17.1, 3.0 Hz, 1H, 3-H), 2.65 (d, *J* = 4.6 Hz, 3H, NCH_3_). ^13^C-NMR: δ 197.41, 167.61, 166.09, 163.48, 163.13, 148.41, 146.93, 131.36, 118.21, 114.56, 112.38, 103.42, 95.84, 94.85, 78.94, 67.49, 56.11, 42.59, 25.82. HRMS (ESI): Calcd. C_19_H_19_NO_7_, [M + H]^+^
*m*/*z*: 374.1234, found: 374.1241.

##### 7-*O*-(2-(Ethylamino)-2-oxoethyl)hesperetin (**4b**)

White crystals, 75% yield, m.p. 171.2–173.6 °C; ^1^H-NMR: δ 12.08 (s, 1H, 5-OH), 9.13 (s, 1H, 3’-OH), 8.14 (t, *J* = 5.5 Hz, 1H, NH), 6.95–6.93 (m, 2H, 2’-H, 5’-H), 6.88 (dd, *J* = 8.4, 1.8 Hz, 1H, 6’-H), 6.12 (d, *J* = 2.2 Hz, 1H, 8-H), 6.10 (d, *J* = 2.2 Hz, 1H, 6-H), 5.49 (dd, *J* = 12.4, 2.9 Hz, 1H, 2-H), 4.53 (s, 2H, ArOCH_2_C=O), 3.78 (s, 3H, OCH_3_), 3.27 (dd, *J* = 17.2, 12.5 Hz, 1H, 3-H), 3.19–3.10 (m, 2H, NCH_2_), 2.76 (dd, *J* = 17.1, 3.0 Hz, 1H, 3-H), 1.04 (t, *J* = 7.2 Hz, 3H, NCCH_3_). ^13^C-NMR: δ 197.39, 166.86, 166.16, 163.48, 163.11, 148.41, 146.93, 131.37, 118.19, 114.55, 112.38, 103.41, 95.82, 94.88, 78.93, 67.48, 56.10, 42.58, 33.71, 15.18. HRMS (ESI): Calcd. C_20_H_21_NO_7_, [M + H]^+^
*m*/*z*: 388.1391, found: 388.1403.

##### 7-*O*-(2-(Dimethylamino)-2-oxoethyl)hesperetin (**4c**)

White crystals, 58% yield, m.p. 164.1–165.8 °C; ^1^H-NMR: δ 12.08 (s, 1H, 5-OH), 9.13 (s, 1H, 3’-OH), 6.95–6.93 (m, 2H, 2’-H, 5’-H), 6.91–6.86 (m, 1H, 6’-H), 6.09 (d, *J* = 2.1 Hz, 1H, 8-H), 6.07 (d, *J* = 2.1 Hz, 1H, 6-H), 5.47 (dd, *J* = 12.6, 2.7 Hz, 1H, 2-H), 4.91 (s, 2H, ArOCH_2_C=O), 3.78 (s, 3H, OCH_3_), 3.26 (dd, *J* = 17.1, 12.7 Hz, 1H, 3-H), 2.95 (s, 3H, NCH_3_), 2.83 (s, 3H, NCH_3_), 2.73 (dd, *J* = 17.1, 2.9 Hz, 1H, 3-H). ^13^C-NMR: δ 197.29, 166.87, 166.71, 163.42, 163.07, 148.41, 146.93, 131.41, 118.24, 114.59, 112.37, 103.16, 95.86, 94.83, 78.96, 66.18, 56.11, 42.62, 35.80, 35.41. HRMS (ESI): Calcd. C_20_H_21_NO_7_, [M + H]^+^
*m*/*z*: 388.1391, found: 388.1401.

##### 7-*O*-(2-(Propylamino)-2-oxoethyl)hesperetin (**4d**)

White powder, 76% yield, m.p. 135.5–137.7 °C; ^1^H-NMR: δ 12.08 (s, 1H, 5-OH), 9.13 (s, 1H, 3’-OH), 8.12 (t, *J* = 5.6 Hz, 1H, NH), 6.95–6.93 (m, 2H, 2’-H, 5’-H), 6.88 (dd, *J* = 8.4, 1.7 Hz, 1H, 6’-H), 6.11 (d, *J* = 2.2 Hz, 1H, 8-H), 6.09 (d, *J* = 2.2 Hz, 1H, 6-H), 5.49 (dd, *J* = 12.4, 2.8 Hz, 1H, 2-H), 4.54 (s, 2H, ArOCH_2_C=O), 3.78 (s, 3H, OCH_3_), 3.27 (dd, *J* = 17.1, 12.5 Hz, 1H, 3-H), 3.08 (dd, *J* = 13.2, 6.6 Hz, 2H, NCH_2_), 2.76 (dd, *J* = 17.1, 3.0 Hz, 1H, 3-H), 1.52–1.36 (m, 2H, NCCH_2_), 0.83 (t, *J* = 7.4 Hz, 3H, NCCCH_3_). ^13^C-NMR: δ 197.38, 167.06, 166.21, 163.48, 163.10, 148.40, 146.93, 131.37, 118.18, 114.55, 112.37, 103.39, 95.80, 94.89, 78.92, 67.46, 56.10, 42.58, 40.57, 22.79, 11.78. HRMS (ESI): Calcd. C_21_H_23_NO_7_, [M + H]^+^
*m*/*z*: 402.1557, found: 402.1562.

##### 7-*O*-(2-(Isopropylamino)-2-oxoethyl)hesperetin (**4e**)

White powder, 73% yield, m.p. 157.6–160.4 °C; ^1^H-NMR: δ 12.08 (s, 1H, 5-OH), 9.13 (s, 1H, 3’-OH), 7.96 (d, *J* = 7.8 Hz, 1H, NH), 6.95–6.93 (m, 2H, 2’-H, 5’-H), 6.88 (dd, *J* = 8.3, 1.8 Hz, 1H, 6’-H), 6.11 (d, *J* = 2.2 Hz, 1H, 8-H), 6.09 (d, *J* = 2.2 Hz, 1H, 6-H), 5.49 (dd, *J* = 12.3, 2.9 Hz, 1H, 2-H), 4.51 (s, 2H, ArOCH_2_C=O), 3.92 (dq, *J* = 13.4, 6.6 Hz, 1H, CHNC=O), 3.78 (s, 3H, OCH_3_), 3.27 (dd, *J* = 17.1, 12.4 Hz, 1H, 3-H), 2.76 (dd, *J* = 17.1, 3.0 Hz, 1H, 3-H), 1.09 (d, *J* = 6.6 Hz, 6H, (CH_3_)_2_CN). ^13^C-NMR: δ 197.36, 166.31, 166.11, 163.49, 163.08, 148.39, 146.93, 131.39, 118.17, 114.54, 112.37, 103.37, 95.77, 94.89, 78.90, 67.44, 56.09, 42.58, 40.85, 22.68, 22.68. HRMS (ESI): Calcd. C_21_H_23_NO_7_, [M + H]^+^
*m*/*z*: 402.1557, found: 402.1562.

##### 7-*O*-(2-(Butylamino)-2-oxoethyl)hesperetin (**4f**)

White powder, 80% yield, m.p. 136.9–138.4 °C; ^1^H-NMR: δ 12.08 (s, 1H, 5-OH), 9.13 (s, 1H, 3’-OH), 8.10 (t, *J* = 5.7 Hz, 1H, NH), 6.94 (dd, *J* = 5.1, 3.2 Hz, 2H, 2’-H, 5’-H), 6.88 (dd, *J* = 8.4, 1.8 Hz, 1H, 6’-H), 6.11 (d, *J* = 2.2 Hz, 1H, 8-H), 6.09 (d, *J* = 2.2 Hz, 1H, 6-H), 5.49 (dd, *J* = 12.4, 2.9 Hz, 1H, 2-H), 4.54 (s, 2H, ArOCH_2_C=O), 3.78 (s, 3H, OCH_3_), 3.26 (dd, *J* = 17.1, 12.5 Hz, 1H, 3-H), 3.11 (dd, *J* = 13.0, 6.7 Hz, 2H, NCH_2_), 2.76 (dd, *J* = 17.1, 3.0 Hz, 1H, 3-H), 1.46–1.34 (m, 2H, NCCH_2_), 1.33–1.18 (m, 2H, NCCCH_2_), 0.86 (t, *J* = 7.3 Hz, 3H, NCCCCH_3_). ^13^C-NMR: δ 197.37, 167.01, 166.21, 163.49, 163.11, 148.40, 146.93, 131.38, 118.17, 114.54, 112.37, 103.38, 95.82, 94.87, 78.94, 67.47, 56.10, 42.60, 38.46, 31.62, 19.97, 14.11. HRMS (ESI): Calcd. C_22_H_25_NO_7_, [M + H]^+^
*m*/*z*: 416.1704, found: 416.1717.

##### 7-*O*-(2-(Isobutylamino)-2-oxoethyl)hesperetin (**4g**)

White powder, 77% yield, m.p. 136.9–139.4 °C; ^1^H-NMR: δ 12.08 (s, 1H, 5-OH), 9.12 (s, 1H, 3’-OH), 8.12 (t, *J* = 5.9 Hz, 1H, NH), 6.97–6.91 (m, 2H, 2’-H, 5’-H), 6.88 (dd, *J* = 8.4, 1.7 Hz, 1H, 6’-H), 6.11 (d, *J* = 2.2 Hz, 1H, 8-H), 6.09 (d, *J* = 2.2 Hz, 1H, 6-H), 5.49 (dd, *J* = 12.4, 2.9 Hz, 1H, 2-H), 4.57 (s, 2H, ArOCH_2_C=O), 3.78 (s, 3H, OCH_3_), 3.26 (dd, *J* = 17.1, 12.5 Hz, 1H, 3-H), 2.94 (t, *J* = 6.4 Hz, 2H, NCH_2_), 2.76 (dd, *J* = 17.1, 3.0 Hz, 1H, 3-H), 1.72 (dp, *J* = 13.4, 6.7 Hz, 1H, NCCHC_2_), 0.83 (d, *J* = 6.7 Hz, 6H, NCC(CH_3_)_2_). ^13^C-NMR: δ 197.37, 167.15, 166.26, 163.49, 163.09, 148.39, 146.93, 131.38, 118.16, 114.54, 112.37, 103.37, 95.78, 94.90, 78.92, 67.45, 56.10, 46.26, 42.59, 28.51, 20.50, 20.50. HRMS (ESI): Calcd. C_22_H_25_NO_7_, [M + H]^+^
*m*/*z*: 416.1704, found: 416.1716.

##### 7-*O*-(2-(tert-Butylamino)-2-oxoethyl)hesperetin (**4h**)

White powder, 53% yield, m.p. 140.5–142.4 °C; ^1^H-NMR: δ 12.08 (s, 1H, 5-OH), 9.13 (s, 1H, 3’-OH), 7.61 (br, 1H, NH), 6.97–6.91 (m, 2H, 2’-H, 5’-H), 6.88 (dd, *J* = 8.4, 1.7 Hz, 1H, 6’-H), 6.08 (d, *J* = 2.2 Hz, 1H, 8-H), 6.06 (d, *J* = 2.2 Hz, 1H, 6-H), 5.49 (dd, *J* = 12.3, 2.9 Hz, 1H, 2-H), 4.48 (s, 2H, ArOCH_2_C=O), 3.78 (s, 3H, OCH_3_), 3.26 (dd, *J* = 17.1, 12.4 Hz, 1H, 3-H), 2.76 (dd, *J* = 17.1, 3.0 Hz, 1H, 3-H), 1.28 (s, 9H, NC(CH_3_)_3_). ^13^C-NMR: δ 197.33, 166.48, 166.41, 163.51, 163.07, 148.39, 146.93, 131.39, 118.18, 114.54, 112.38, 103.31, 95.61, 94.84, 78.89, 67.51, 56.09, 50.89, 42.57, 28.87, 28.87, 28.87. HRMS (ESI): Calcd. C_22_H_25_NO_7_, [M + H]^+^
*m*/*z*: 416.1704, found: 416.1717.

##### 7-*O*-(2-(Diethylamino)-2-oxoethyl)hesperetin (**4i**)

White powder, 46% yield, m.p. 146.3–147.8 °C; ^1^H-NMR: δ 12.08 (s, 1H, 5-OH), 9.13 (s, 1H, 3’-OH), 6.97–6.91 (m, 2H, 2’-H, 5’-H), 6.89 (d, *J* = 8.2 Hz, 1H, 6’-H), 6.08 (d, *J* = 1.9 Hz, 1H, 8-H), 6.05 (d, *J* = 1.9 Hz, 1H, 6-H), 5.48 (dd, *J* = 12.5, 2.4 Hz, 1H, 2-H), 4.88 (s, 2H, ArOCH_2_C=O), 3.78 (s, 3H, OCH_3_), 3.34–3.20 (m, 5H, N(CH_2_C)_2_, 3-H), 2.73 (dd, *J* = 17.1, 2.7 Hz, 1H, 3-H), 1.13 (t, *J* = 7.0 Hz, 3H, NCCH_3_), 1.02 (t, *J* = 7.0 Hz, 3H, NCCH_3_). ^13^C-NMR: δ 197.29, 166.83, 165.82, 163.45, 163.08, 148.41, 146.93, 131.40, 118.23, 114.58, 112.37, 103.19, 95.78, 94.85, 79.65, 78.96, 66.38, 56.10, 42.62, 40.86, 14.53, 13.33. HRMS (ESI): Calcd. C_22_H_25_NO_7_, [M + H]^+^
*m*/*z*: 416.1704, found: 416.1714.

##### 7-*O*-(2-(Cyclopropylamino)-2-oxoethyl)hesperetin (**4j**)

White powder, 66% yield, m.p. 190.8–193.1 °C; ^1^H-NMR:) δ 12.08 (s, 1H, 5-OH), 9.13 (s, 1H, 3’-OH), 8.17 (d, *J* = 4.0 Hz, 1H, NH), 6.95–6.93 (m, 2H, 2’-H, 5’-H), 6.88 (dd, *J* = 8.4, 1.7 Hz, 1H, 6’-H), 6.10 (d, *J* = 2.2 Hz, 1H, 8-H), 6.08 (d, *J* = 2.2 Hz, 1H, 6-H), 5.49 (dd, *J* = 12.4, 2.8 Hz, 1H, 2-H), 4.51 (s, 2H, ArOCH_2_C=O), 3.78 (s, 3H, OCH_3_), 3.27 (dd, *J* = 17.1, 12.5 Hz, 1H, 3-H), 2.76 (dd, *J* = 17.1, 3.0 Hz, 1H, 3-H), 2.72–2.64 (m, 1H, NCH), 0.68–0.60 (m, 2H, NCCH_2_), 0.51–0.43 (m, 2H, NCCH_2_). ^13^C-NMR: δ 197.38, 168.33, 166.24, 163.46, 163.09, 148.40, 146.93, 131.37, 118.19, 114.55, 112.37, 103.38, 95.78, 94.84, 78.92, 67.37, 56.09, 42.58, 22.64, 6.05, 6.05. HRMS (ESI): Calcd. C_21_H_21_NO_7_, [M + H]^+^
*m*/*z*: 400.1401, found: 400.1407.

##### 7-*O*-(2-(Cyclopentylamino)-2-oxoethyl)hesperetin (**4k**)

White powder, 79% yield, m.p. 187.5–190.0 °C; ^1^H-NMR: δ 12.08 (s, 1H, 5-OH), 9.13 (s, 1H, 3’-OH), 8.04 (d, *J* = 7.4 Hz, 1H, NH), 6.95–6.93 (m, 2H, 2’-H, 5’-H), 6.88 (dd, *J* = 8.4, 1.7 Hz, 1H, 6’-H), 6.10 (d, *J* = 2.2 Hz, 1H, 8-H), 6.08 (d, *J* = 2.2 Hz, 1H, 6-H), 5.49 (dd, *J* = 12.3, 2.8 Hz, 1H, 2-H), 4.52 (s, 2H, ArOCH_2_C=O), 4.12–3.99 (m, 1H, NCH), 3.78 (s, 3H, OCH_3_), 3.26 (dd, *J* = 17.1, 12.4 Hz, 1H, 3-H), 2.76 (dd, *J* = 17.1, 3.0 Hz, 1H, 3-H), 1.80 (dt, *J* = 12.6, 6.1 Hz, 2H, NCCH_2_), 1.72–1.58 (m, 2H, NCCH_2_), 1.56–1.46 (m, 2H, NCCCH_2_), 1.46–1.34 (m, 2H, NCCCH_2_). ^13^C-NMR: δ 197.35, 166.54, 166.37, 163.49, 163.08, 148.39, 146.93, 131.39, 118.17, 114.54, 112.36, 103.34, 95.73, 94.87, 78.90, 67.42, 56.09, 50.63, 42.58, 32.54, 32.54, 23.92, 23.92. HRMS (ESI): Calcd. C_23_H_25_NO_7_, [M + H]^+^
*m*/*z*: 428.1714, found: 428.1723.

##### 7-*O*-(2-(Cyclohexylamino)-2-oxoethyl)hesperetin (**4l**)

White powder, 80% yield, m.p. 181.9–184.1 °C; ^1^H-NMR: δ 12.08 (s, 1H, 5-OH), 9.13 (s, 1H, 3’-OH), 7.95 (d, *J* = 8.0 Hz, 1H, NH), 6.95–6.93 (m, 2H, 2’-H, 5’-H), 6.88 (dd, *J* = 8.4, 1.7 Hz, 1H, 6’-H), 6.11 (d, *J* = 2.1 Hz, 1H, 8-H), 6.09 (d, *J* = 2.2 Hz, 1H, 6-H), 5.49 (dd, *J* = 12.3, 2.8 Hz, 1H, 2-H), 4.53 (s, 2H, ArOCH_2_C=O), 3.78 (s, 3H, OCH_3_), 3.60 (m, 1H, NCH), 3.26 (dd, *J* = 17.1, 12.4 Hz, 1H, 3-H), 2.76 (dd, *J* = 17.1, 3.0 Hz, 1H, 3-H), 1.78–1.62 (m, 4H, NC(CH_2_)_2_), 1.57–1.54 (m, 1H, NCCCCH), 1.34–1.18 (m, 4H, NCC(CH_2_)_2_), 1.15–1.09 (m, 1H, NCCCCH). ^13^C-NMR: δ 197.35, 166.33, 166.07, 163.49, 163.07, 148.39, 146.93, 131.39, 118.16, 114.54, 112.36, 103.35, 95.74, 94.89, 78.90, 67.42, 56.09, 47.96, 42.58, 32.71, 32.71, 25.61, 25.07, 25.07. HRMS (ESI): Calcd. C_24_H_27_NO_7_, [M + H]^+^
*m*/*z*: 442.1860, found: 442.1870.

##### 7-*O*-(2-(Pyrrolidin-1-yl)-2-oxoethyl)hesperetin (**5a**)

White powder, 74% yield, m.p. 127.1–130.5 °C; ^1^H-NMR: δ 12.08 (s, 1H, 5-OH), 9.13 (s, 1H, 3’-OH), 6.95–6.93 (m, 2H, 2’-H, 5’-H), 6.89 (dd, *J* = 8.4, 1.8 Hz, 1H, 6’-H), 6.09 (d, *J* = 2.3 Hz, 1H, 8-H), 6.07 (d, *J* = 2.3 Hz, 1H, 6-H), 5.47 (dd, *J* = 12.7, 2.8 Hz, 1H, 2-H), 4.81 (s, 2H, ArOCH_2_C=O), 3.78 (s, 3H, OCH_3_), 3.42 (t, *J* = 6.8 Hz, 2H, CH_2_NC=O), 3.30 (t, *J* = 6.8 Hz, 2H, CH_2_NC=O), 3.26 ((dd, *J* = 17.1, 12.4 Hz, 1H, 3-H), 2.72 (dd, *J* = 17.1, 3.0 Hz, 1H, 3-H), 1.88 (p, *J* = 6.7 Hz, 2H, CH_2_CNC=O), 1.81–1.71 (p, *J* = 6.7 Hz, 2H, CH_2_CNC=O). ^13^C-NMR: δ 197.30, 166.85, 165.17, 163.42, 163.08, 148.42, 146.92, 131.40, 118.26, 114.59, 112.36, 103.18, 95.85, 94.79, 78.98, 66.59, 56.11, 46.03, 44.93, 42.63, 26.07, 23.97. HRMS (ESI): Calcd. C_22_H_23_NO_7_, [M + H]^+^
*m*/*z*: 414.1557, found: 414.1563.

##### 7-*O*-(2-(Thiazolidin-3-yl)-2-oxoethyl)hesperetin (**5b**)

White crystals, 76% yield, m.p. 152.4–154.7 °C; ^1^H-NMR: δ 12.08 (s, 1H, 5-OH), 9.13 (s, 1H, 3’-OH), 6.95–6.93 (m, 2H, 2’-H, 5’-H), 6.89 (dd, *J* = 8.3, 1H, 6’-H), 6.13 (d, *J* = 2.2 Hz, 1H, 8-H), 6.10 (d, *J* = 2.2 Hz, 1H, 6-H), 5.48 (dd, *J* = 12.6, 2.8 Hz, 1H, 2-H), 4.94 (s, 2H, ArOCH_2_C=O), 4.52 (br, 2H, SCH_2_N), 3.78 (s, 3H, OCH_3_), 3.72 (t, *J* = 6.2 Hz, 1H, NCHCS), 3.67 (t, *J* = 6.3 Hz, 1H, NCHCS)), 3.27 (dd, *J* = 17.1, 12.7 Hz, 1H, 3-H), 3.11 (t, *J* = 6.2 Hz, 1H, NCCHS), 2.98 (t, *J* = 6.3 Hz, 1H, NCCHS), 2.73 (dd, *J* = 17.1, 3.0 Hz, 1H, 3-H). ^13^C-NMR: δ 197.34, 166.67, 165.39, 163.43, 163.11, 148.42, 146.92, 131.38, 118.26, 114.59, 112.36, 103.25, 95.87, 94.83, 78.98, 66.61, 56.11, 48.59, 47.54, 42.62, 31.25. HRMS (ESI): Calcd. C_21_H_21_NO_7_S, [M + H]^+^
*m*/*z*: 432.1111, found: 432.1126.

##### 7-*O*-(2-(piperidin-1-yl)-2-oxoethyl)hesperetin (**6a**)

White powder, 78% yield, m.p. 165.4–167.9 °C; ^1^H-NMR: δ 12.08 (s, 1H, 5-OH), 9.13 (s, 1H, 3’-OH), 6.94 (d, *J* = 7.1 Hz, 2H, 2’-H, 5’-H), 6.89 (d, *J* = 8.2 Hz, 1H, 6’-H), 6.08 (d, *J* = 2.1 Hz, 1H, 8-H), 6.06 (d, *J* = 2.1 Hz, 1H, 6-H), 5.48 (dd, *J* = 12.3, 2.1 Hz, 1H, 2-H), 4.90 (s, 2H, ArOCH_2_C=O), 3.78 (s, 3H, OCH_3_), 3.41 (m, 2H, NCH_2_), 3.34 (m, 2H, NCH_2_), 3.26 (dd, *J* = 17.0, 12.8 Hz, 1H, 3-H), 2.73 (dd, *J* = 17.0, 2.4 Hz, 1H, 3-H), 1.70–1.34 (m, 6H, NC(CH_2_)_2_CH_2_). ^13^C-NMR: δ 197.30, 166.83, 165.01, 163.44, 163.08, 148.41, 146.93, 131.41, 118.23, 114.58, 112.36, 103.18, 95.81, 94.83, 78.96, 66.47, 56.10, 45.42, 42.62, 42.62, 26.34, 25.72, 24.37. HRMS (ESI): Calcd. C_23_H_25_NO_7_, [M + H]^+^
*m*/*z*: 428.1704, found: 428.1718.

##### 7-*O*-(2-(3-(Hydroxymethyl)piperidin-1-yl)-2-oxoethyl)hesperetin (**6b**)

White powder, 40% yield, m.p. 168.6–171.8 °C; ^1^H-NMR: δ 12.08 (s, 1H, 5-OH), 9.13 (s, 1H, 3’-OH), 6.97–6.91 (m, 2H, 2’-H, 5’-H), 6.89 (dd, *J* = 8.3, 1.6 Hz, 1H, 6’-H), 6.08 (d, *J* = 2.3 Hz, 1H, 8-H), 6.06 (d, *J* = 2.3 Hz, 1H, 6-H), 5.48 (dd, *J* = 12.5, 2.5 Hz, 1H, 2-H), 4.91 (s, 2H, ArOCH_2_C=O), 4.61–4.55 (m, 1H, OH), 3.78 (s, 3H, OCH_3_), 3.66 (d, *J* = 13.0 Hz, 1H, CH_2_OH), 3.36 (m, 1H, CH_2_OH), 3.29–3.12 (m, 3H, 3-H, NCH_2_), 3.01–2.77 (m, 2H, NCH_2_), 2.73 (dd, *J* = 17.1, 2.8 Hz, 1H, 3-H), 1.79–1.50 (m, 3H, CHCH_2_CH_2_), 1.44–1.38 (m, 1H, CHCH_2_CH_2_), 1.34–1.19 (m, 1H, CHCH_2_CH_2_). ^13^C-NMR: δ 197.29, 166.82, 165.09, 163.44, 163.07, 148.41, 146.93, 131.41, 118.24, 114.58, 112.37, 103.19, 95.80, 94.84, 78.96, 66.54, 64.11, 56.10, 47.60, 45.40, 42.63, 39.08, 27.23, 25.40. HR-MS (ESI): Calcd. C_24_H_27_NO_8_, [M + H]^+^ m/z: 458.1819, found: 458.1828.

##### 7-*O*-(2-(2-(Hydroxymethyl)piperidin-1-yl)-2-oxoethyl)hesperetin (**6c**)

White powder, 36% yield, m.p. 123.2–125.8 °C; ^1^H-NMR: δ 12.08 (s, 1H, 5-OH), 9.13 (s, 1H, 3’-OH), 6.97–6.92 (m, 2H, 2’-H, 5’-H), 6.89 (dd, *J* = 8.4, 1.7 Hz, 1H, 6’-H), 6.07 (d, *J* = 15.6 Hz, 2H, 8-H, 6-H), 5.48 (dd, *J* = 12.6, 2.7 Hz, 1H, 2-H), 4.92 (s, 2H, ArOCH_2_C=O), 4.24 (d, *J* = 12.9 Hz, 1H, OH), 3.91–3.71 (m, 4H, OCH_3_, CH_2_OH), 3.55–3.48 (m, 2H, NCH, CH_2_OH), 3.45–3.37 (m, 2H, NCH_2_), 3.26 (dd, *J* = 17.1, 12.7 Hz, 1H, 3-H), 2.72 (dd, *J* = 17.0 Hz, 2.7 Hz, 1H, 3-H), 1.70–1.34 (m, 5H, CH_2_CH_2_CH_2_), 1.31–1.18 (m, 1H, CH_2_CH_2_CH_2_). ^13^C-NMR: δ 197.25, 167.12, 166.26, 163.44, 163.05, 148.42, 146.93, 131.45, 118.26, 114.60, 112.38, 103.07, 95.82, 94.82, 78.96, 66.42, 59.45, 56.11, 53.35, 42.65, 36.56, 25.92, 25.43, 21.22. HR-MS (ESI): Calcd. C_24_H_27_NO_8_, [M + H]^+^
*m*/*z*: 458.1819, found: 458.1827.

##### 7-*O*-(2-(4-Methylpiperazin-1-yl)-2-oxoethyl)hesperetin (**6d**)

White crystals, 56% yield, m.p. 167.3–170.1 °C; ^1^H-NMR: δ 12.08 (s, 1H, 5-OH), 9.14 (s, 1H, 3’-OH), 6.97–6.91 (m, 2H, 2’-H, 5’-H), 6.89 (dd, *J* = 8.3, 1.8 Hz, 1H, 6’-H), 6.10 (d, *J* = 2.2 Hz, 1H, 8-H), 6.07 (d, *J* = 2.2 Hz, 1H, 6-H), 5.48 (dd, *J* = 12.6, 2.8 Hz, 1H, 2-H), 4.92 (s, 2H, ArOCH_2_C=O), 3.78 (s, 3H, OCH_3_), 3.43 (m, 2H, NCCH_2_NC=O), 3.39 (m, 2H, NCCH_2_NC=O), 3.26 (dd, *J* = 17.1, 12.7 Hz, 1H, 3-H), 2.73 (dd, *J* = 17.1, 3.0 Hz, 1H, 3-H), 2.32 (m, 2H, NCH_2_CNC=O), 2.26 (m, 2H, NCH_2_CNC=O), 2.17 (s, 3H, NCH_3_). ^13^C-NMR: δ 197.30, 166.76, 165.37, 163.44, 163.08, 148.41, 146.93, 131.40, 118.23, 114.58, 112.36, 103.20, 95.83, 94.84, 78.96, 66.30, 56.10, 54.99, 54.67, 46.12, 44.28, 42.62, 41.61. HRMS (ESI): Calcd. C_23_H_26_N_2_O_7_, [M + H]^+^
*m*/*z*: 443.1823, found: 443.1833.

##### 7-*O*-(2-(4-Ethylpiperazin-1-yl)-2-oxoethyl)hesperetin (**6e**)

White crystals, 58% yield, m.p. 160.7–162.9 °C; ^1^H-NMR: δ 12.08 (s, 1H, 5-OH), 9.14 (s, 1H, 3’-OH), 6.96–6.91 (m, 2H, 2’-H, 5’-H), 6.89 (d, *J* = 8.2 Hz, 1H, 6’-H), 6.10 (d, *J* = 1.7 Hz, 1H, 8-H), 6.07 (d, *J* = 1.9 Hz, 1H, 6-H), 5.47 (dd, *J* = 12.6, 2.6 Hz, 1H, 2-H), 4.92 (s, 2H, ArOCH_2_C=O), 3.78 (s, 3H, OCH_3_), 3.43 (m, 2H, NCCH_2_NC=O), 3.39 (m, 2H, NCCH_2_NC=O), 3.26 (dd, *J* = 17.1, 12.7 Hz, 1H, 3-H), 2.73 (dd, *J* = 17.1, 2.7 Hz, 1H, 3-H), 2.42–2.26 (m, 6H, N(CH_2_C)_2_NC=O, NCH_2_C), 0.99 (t, *J* = 7.1 Hz, 3H, NCCH_3_). ^13^C-NMR: δ 197.30, 166.77, 165.32, 163.44, 163.08, 148.41, 146.93, 131.40, 118.23, 114.58, 112.36, 103.20, 95.83, 94.85, 78.96, 66.31, 56.10, 52.85, 52.40, 51.97, 44.41, 42.63, 41.73, 12.33. HRMS (ESI): Calcd. C_24_H_28_N_2_O_7_, [M + H]^+^
*m*/*z*: 457.1979, found: 457.1985.

##### 7-*O*-(2-Morpholino-2-oxoethyl)hesperetin (**6f**)

White powder, 73% yield, m.p. 188.7–189.3 °C; ^1^H-NMR: δ 12.08 (s, 1H, 5-OH), 9.13 (s, 1H, 3’-OH), 6.96–6.92 (m, 2H, 2’-H, 5’-H), 6.89 (dd, *J* = 8.4, 1.7 Hz, 1H, 6’-H), 6.11 (d, *J* = 2.2 Hz, 1H, 8-H)), 6.09 (d, *J* = 2.2 Hz, 1H, 6-H)), 5.48 (dd, *J* = 12.6, 2.7 Hz, 1H, 2-H)), 4.94 (s, 2H, ArOCH_2_C=O), 3.78 (s, 3H, OCH_3_), 3.63–3.52 (m, 4H, O(CH_2_C)_2_N), 3.42 (m, 4H, O(CCH_2_)_2_N), 3.27 (dd, *J* = 17.1, 12.7 Hz, 1H, 3-H), 2.73 (dd, *J* = 17.1, 2.9 Hz, 1H, 3-H). ^13^C-NMR: δ 197.32, 166.75, 165.65, 163.44, 163.10, 148.42, 146.92, 131.40, 118.25, 114.59, 112.36, 103.22, 95.86, 94.85, 78.98, 66.49, 66.37, 66.21, 56.10, 44.92, 42.62, 42.01. HR-MS (ESI): Calcd. C_22_H_23_NO_8_, [M + H]^+^
*m*/*z*: 430.1506, found: 430.1511.

##### 7-*O*-(2-((2-Hydroxyethyl)amino)-2-oxoethyl)hesperetin (**7a**)

White crystals, 40% yield, m.p. 183.8–185.7 °C; ^1^H-NMR: δ 12.08 (s, 1H, 5-OH), 9.13 (s, 1H, 3’-OH), 8.09 (t, *J* = 5.6 Hz, 1H, NH), 6.95–6.93 (m, 2H, 2’-H, 5’-H), 6.89 (dd, *J* = 8.4, 1.8 Hz, 1H, 6’-H), 6.12 (d, *J* = 2.2 Hz, 1H, 8-H), 6.10 (d, *J* = 2.2 Hz, 1H, 6-H), 5.49 (dd, *J* = 12.4, 2.9 Hz, 1H, 2-H), 4.74 (t, *J* = 5.5 Hz, 1H, OH), 4.55 (s, 2H, ArOCH_2_C=O), 3.78 (s, 3H, OCH_3_), 3.43 (q, *J* = 5.9 Hz, 2H, NCCH_2_O), 3.27 (dd, *J* = 17.2, 12.5 Hz, 1H, 3-H), 3.20 (q, *J* = 6.0 Hz, 2H, NCH_2_CO), 2.76 (dd, *J* = 17.1, 3.0 Hz, 1H, 3-H). ^13^C-NMR: δ 197.38, 167.29, 166.16, 163.49, 163.12, 148.41, 146.93, 131.37, 118.21, 114.56, 112.40, 103.42, 95.82, 94.87, 78.93, 67.40, 60.06, 56.11, 42.58, 41.71. HR-MS (ESI): Calcd. C_20_H_21_NO_8_, [M + H]^+^
*m*/*z*: 404.1360, found: 404.1369.

##### 7-*O*-(2-((2-(Dimethylamino)ethyl)amino)-2-oxoethyl)hesperetin (**7b**)

White crystals, 52% yield, m.p. 137.1–139.4 °C; ^1^H-NMR: δ 12.08 (s, 1H, 5-OH), 9.15 (s, 1H, 3’-OH), 8.04 (t, *J* = 5.6 Hz, 1H, NH), 6.95–6.93 (m, 2H, 2’-H, 5’-H), 6.88 (dd, *J* = 8.4, 1.7 Hz, 1H, 6’-H), 6.11 (d, *J* = 2.2 Hz, 1H, 8-H), 6.09 (d, *J* = 2.2 Hz, 1H, 6-H), 5.49 (dd, *J* = 12.4, 2.8 Hz, 1H, 2-H), 4.55 (s, 2H, ArOCH_2_C=O), 3.78 (s, 3H, OCH_3_), 3.32–3.16 (m, 3H, 3-H, NCCH_2_NC=O), 2.76 (dd, *J* = 17.1, 3.0 Hz, 1H, 3-H), 2.30 (t, *J* = 6.6 Hz, 2H, NCH_2_CNC=O), 2.13 (s, 6H, (CH_3_)_2_N). ^13^C-NMR: δ 197.25, 167.12, 166.27, 163.44, 163.05, 148.42, 146.93, 131.45, 118.26, 114.60, 112.38, 103.07, 95.82, 94.80, 78.96, 66.42, 60.24, 59.45, 56.11, 53.35, 42.65, 36.56. HRMS (ESI): Calcd. C_22_H_26_N_2_O_7_, [M + H]^+^
*m*/*z*: 431.1823, found: 431.1830.

##### 7-*O*-(2-((2-Mmorpholinoethyl)amino)-2-oxoethyl) hesperetin (**7c**)

White crystals, 54% yield, m.p. 160.7–162.9 °C; ^1^H-NMR: δ 12.08 (s, 1H, 5-OH), 9.13 (s, 1H, 3’-OH), 8.02 (t, *J* = 5.6 Hz, 1H, NH), 6.95–6.93 (m, 2H, 2’-H, 5’-H), 6.88 (dd, *J* = 8.3, 1.8 Hz, 1H, 6’-H), 6.11 (d, *J* = 2.1 Hz, 1H, 8-H), 6.10 (d, *J* = 2.1 Hz, 1H, 6-H), 5.48 (dd, *J* = 12.4, 2.8 Hz, 1H, 2-H), 4.56 (s, 2H, ArOCH_2_C=O), 3.78 (s, 3H, OCH_3_), 3.58–3.46 (m, 4H, (OCH_2_CN)_2_), 3.33–3.18 (m, 3H, NCCH_2_NC=O, 3-H), 2.76 (dd, *J* = 17.1, 3.0 Hz, 1H, 3-H), 2.37–2.34 (m, *J* = 6.5 Hz, 6H, (OCCH_2_N)_2_, NCH_2_CNC=O). ^13^C-NMR: δ 197.39, 167.16, 166.13, 163.50, 163.14, 148.42, 146.93, 131.35, 118.20, 114.56, 112.37, 103.43, 95.86, 94.87, 78.98, 67.45, 66.61, 66.61, 57.62, 56.10, 53.65, 42.60, 35.98. HRMS (ESI): Calcd. C_24_H_28_N_2_O_8_, [M + H]^+^
*m*/*z*: 473.1928, found: 473.1932.

HRMS, ^1^H-NMR and ^13^C-NMR of the compounds are available in Appendix A.

### 3.2. Biological Assays

#### 3.2.1. Cell Culture

The RAW264.7 cell line was purchased from Cell Bank of Chinese Academy of Sciences (Shanghai, China). The cells were cultured in DMEM (Hyclone, Logan, UT, USA), supplemented with 10% fetal bovine serum (Biological industries) and antibiotics (100 U/mL penicillin A and 100 U/mL streptomycin), and maintained at 37 °C in a humidified atmosphere containing 5% CO_2_ [31].

#### 3.2.2. Determination of Cell Viability

Cytotoxicity and cell viability was evaluated by a MTT (Sigma, Shanghai, China) assay using a method similar to the one described above [25]. Cells were inoculated at a density of 7.0 × 10^3^ cells/well into 96-well plate and cultured at 37 °C for 12 h. Culture cells treated with the same one-thousandth of the DMSO solvent as the vehicle (control) and 40 μM compounds for 24 h. After that, MTT dissolved in phosphate buffered saline (PBS) and was added to the culture medium to reach a final concentration of 0.5 mg/mL and incubated for another 4 h. The media was removed and 200 μL of DMSO was added and incubated for 30 min. Finally, the cell viability (%) was calculated based on the absorbance measured at the wavelength of 490 nm and measured using a microplate reader (Synergy HTX, Biotek, Winowski, VT, USA). The cell viability were calculated by GraphPad Prism 6 (GraphPad, San Diego, CA, USA).

#### 3.2.3. Assessment of Nitric Oxide (NO)

Nitrite levels were determined using the Griess method for NO has a short half-life and is oxidized to stable nitrite. RAW 264.7 cells were plated at a density of 2 × 10^5^ per well in a 24-well culture plate and incubated in a 37 °C humidified incubator (5% CO_2_) [32]. After overnight Cultured, discarded the supernatant, added pre-configured compounds of different concentrations to the 24-well plate, and added the same one-thousandth of the DMSO solvent as the control to the experimental group for 1 h. Then LPS was added to experimental and model groups at a concentration of 1 mg/mL for 24 h compared to vehicle control. Following treatment, media was collected and centrifuged at 2000 rpm for 5 min to remove cellular debris. The medium from treated cells was removed and placed into a 96-well plate (50 µL per well) and was measured in culture supernatant using Griess reagent at 540 nm. The IC_50_ values of compounds were calculated by SPSS23.

#### 3.2.4. Measurement of Cytokines

RAW264.7 cells (2 × 10^5^ cells/well) were cultured in 24-well plate and pretreated with 10 μM of compounds for 1 h. And then stimulated with LPS, 1 mg/mL, for 24 h. The cell culture supernatant were collected and the levels of TNF-α, IL-6 and IL-1β were determined using the ELISA kit (Elabscience, Wuhan, China). The optical density (OD) was measured at 540 nm [33]. The results were analyzed using Origin Pro 8 (OriginLab, Northampton, MA, USA).

#### 3.2.5. Western Blotting Analysis

After the treatment with LPS (1 mg/mL) in the presence different concentration gradient or absence of compounds, removed culture supernatants, and cells were collected and lysed with RIPA lysis buffer (Beyotime, Shanghai, China) and phosphatase inhibitors (Beibokit, Bestbio, Shanghai, China). After centrifugation (4 °C, 30 min, 12,000 rpm, Beckman Coulter, Inc., Fullerton, CA, USA), the total protein concentration was determined using BCA Protein Assay Kit (Beyotime, Shanghai, China). Lysates containing 30 mg of total proteins were fractionated on SDS polyacrylamide gel and transferred to a polyvinylidene difluoride membranes (PVDF) (Merck Millipore, Tullagreen Carrigtwohill, Ireland). Membranes were blocked with 5% skim milk or 3% BSA (Albumin Bovine V, Sigma) in TBST and incubated at 4 °C overnight with their respective primary antibody: β-actin (1:1000 dilution; Abcam, Cambridge, MA, USA), iNOS (1:1000 dilution; Abcam, Cambridge, MA, USA), COX-2 (1:1000 dilution; CST, Danvers, MA, USA ), NF-κB (1:1000 dilution; CST, Danvers, MA, USA). After washing three times with TBST, membranes were incubated with secondary antibody for 1 h in room temperature on the shaker. Protein bands were visualized by the ultra-sensitive enhanced chemiluminescent (ECL) substrate (ECL-plus, Thermo Scientific, Rockford, IL, USA), after washing three times with TBST [34].

## 4. Statistical Analysis

All experiments were repeated at least three times. Data are presented as the means ± standard deviation (SD) for the indicated number of independently performed experiments. The significance of differences between drug-treated groups compared to vehicle was determined by one way ANOVA using SPSS 23 (IBM, Chicago, IL, USA). The difference was considered statistically significant when *p* (*) < 0.05, *p* (**) < 0.01, *p* (***) < 0.001 or no significance (ns).

## 5. Conclusions

In conclusion, based on their structural characteristics, a series of 7-O-amide hesperetin derivatives were designed and synthesized as potential anti-inflammatory agents. In vitro bioassays showed that title compounds could be alleviated the increase of NO release in LPS-induced RAW264.7 cells. The 7-O-amide side chains were buried in a medium-sized hydrophobic cavity of the bound receptor, while with a suitably extended length of the amide side chain the inhibitory activities were increased. Compounds **4d** and **4k** exhibited more potent inhibitory activity towards NO, IL-6, IL-1β and TNF-α than celecoxib and indomethacin. Further, western blotting showed that compound **4d** significantly decreased the expression levels of iNOS and COX-2, as well as activation of NF-κB in LPS-stimulated RAW264.7 cells. Taken together, compound **4d** exerted its anti-inflammatory activity through inhibition of NO generation as a result of inhibiting NF-κB signalling pathways. These results could be very useful for a SAR study in the future, when we will proceed with modification of the hesperetin structure at the 3’-OH, 5-OH and 4-carbonyl positions.

## 6. Patents

A patent application resulting from the work reported in this manuscript has been submitted to the National Intellectual Property Administration (PRC). The patent application number is CN201910754986.4.

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
