# Peer review of "Design, Synthesis and Investigation of the Potential Anti-Inflammatory Activity of 7-O-Amide Hesperetin Derivatives"

_molecules, 2019, doi:10.3390/molecules24203663_

Round 1
Reviewer 1 Report
(1) Multiple grammatical errors. For instance, Page 1, Line 15: a series of ... was ... Authors' responsibility for an error-free manuscript.
(2) Any abbreviations should be spelled in full at their initial use. For instance, LPS.
(3) Why none of the studies included hesperidin or hesperetin as control? Particularly, cell viability assays.
Reviewer 2 Report
At the first appearance in the text, the abbreviations EDC, HOBT, LPS, MTT, RAW264.7 cells etc. should be explained. The titles of cap.2.2., 2.2.3., 2.2.4., 2.2.5. must be reformulated.
Lines 22: authors should explain what hydrophobic cavity is mentioned; also in the line 50 etc.
Line 20: cytokines instead of cytokine.
Line 31: the Latin names must be written in italics.
Line 33: anti-inflammatory instead of anti-inflammation.
Lines 34-35: the information must be rephrased.
Line 43: in vivo and in vitro must be written in italics; recheck the entire manuscript.
Lines 48-58: this part of cap. Introduction should be reformulated. For example it is necessary a bibliographic reference at the end of the sentence from the line 49. Also, in the introduction section, the authors should mention the objectives of the research and support the study aim based on the literature existing studies, and not to mention the results from the present study, eg.. the most active compound 4d, before the presentation of the Result and Discussion section.
In the Scheme 1, the compounds the compounds 4, 5, 6, 7 must be mentioned.
Lines 78-80: the sentence must be rephrased; the last sentence must be rechecked for spelling errors.
Line 82: the negative control data must be added to Table 1; it is recommended to determine the cells cytotoxicity in the presence of LPS and DMSO (as it was used in the method described on the Materials and Methods section); the results must be expressed as (%), compared to LPS-stimulated cells.
Line 85: the results must be compared to LPS-stimulated cells.
Line 86: recheck the Figure 1 title.
Line 88: authors should rephrase: The difference was considered statistically significant when p(*) < 0.05, p(**) < 0.01, p(***) < 0.001, p (****)< 0.0001 or without statistically significance (ns).; verify the entire manuscript.
Line 95: in the Methods described at Materials and Method section, is given 24 h not 22 h.
Line 99: the reference 24 it is not related to the subject; please reconsider it.
Line 105: compounds instead of compound.
Line 108: reconsider the expression ”As a whole”.
Line 110: Table 2 should contain also the data from negative control, as the results are compared to this group.
Line 112: the sentence is incomplete.
Line 118-120: the information must be rephrased.
Line 120: Figure 2A and Figure 2B instead of Fig.2A and Fig.2B; it is also available for Fig.2C and 2D (line 125).
Line 128: mL instead of ml (it is available for the entire manuscript).
Line 156: recheck the spelling for all paragraph.
Line 167: in the P65 (compared to control) graphic the statistical significance is missing.
Line 175: re-check Brucker AV-400 or AV-400 MHz information.
Cap. Chemistry: The chemical names of compounds should be verified and completed for all of synthesized compounds. For all the compounds the melting points are missing.
Line 183: hesperetin instead of hespertin, also at line 190.
For the compound 1, must be pointed out the chemical shifts for 7-OH and 3’-OH. It is available for the entire manuscript to check the spectral data.
Line 215: the authors should reconsider the information “Substituendum derivatives…”.
Line 227: General procedure for the synthesis of compounds 4a-l, 5a-b, 6a-g, 7a-c, instead of “General procedure for the synthesis of (…)”.
Line 451: the author must improve the method described because it cannot be reproducible; also, the vehicle (control) composition must be specified and the concentration of DMSO.
Line 461: the author must improve the method described because it cannot be reproducible; the vehicle control and model control must be detailed.
Line 471: reconsider the name of the subtitle and verify the spelling from the entire paragraph; Also, IL-1β results are not presented in the manuscript and in this method is specified that the levels of IL-1β were determined.
Line 485: COX-2 must be added at primary antibody.
Line 505: the authors should reconsider the information “compounds 4d reduced decreased”.
In cap. Conclusions is recommended to point out also the importance of COX-2 inhibition for the anti-inflammatory effect.
Reviewer 3 Report
Manuscript presented for opinion describes synthesis of hesperetin derivatives. There is only little novelty in this paper because subject is highly elaborated by authors since some time ego especially concerning new amides as derivatives. Main parts of synthesis are based on published already results. Biological activity repeat also same procedures as it was presented in earlier publications.
Compounds obtained are new compounds, correctly described by spectral methods. Their biological activity was established only on the cellular level. It should be extended with animal models.
Language require improvement, should be checked by native speaker.
At that moment I do not recommend publication. Manuscript would require major revision or should be rejected.
Reviewer 4 Report
Page 1,
Line 19, the extra-space could be deleted
Line 38 in vivo must be in cursive, as well as in vitro and in vivo in line 43
Page 2
In the chemistry section the authors do not explain the selectivity of the hydrolysis reaction to get there -OH group instead of four groups in compound 1.
Page 3
Line 71, Evaluation of biological evaluation, no not have sense.
Line 84, The authors do not gave the meaning or the interpretation of significance on the statistic treatment (table 1, table 2 and figure 1 on page 4).
On the figure 3 is iNOS instead of INOS
Page 8
Line 153, 4d must be in bold form.
Page 16
Line 479, an extra space must be deleted.
